# Molecular Dynamics as a Means to Investigate Grain Size and Strain Rate Effect on Plastic Deformation of 316 L Nanocrystalline Stainless-Steel

**DOI:** 10.3390/ma13143223

**Published:** 2020-07-20

**Authors:** Abdelrahim Husain, Peiqing La, Yue Hongzheng, Sheng Jie

**Affiliations:** 1State Key Laboratory of Advanced Processing and Recycling of Nonferrous Metals, Lanzhou University of Technology, Lanzhou 730050, China; abdosh.husian@yahoo.com (A.H.); zhengyuehong1986@126.com (Y.H.); Shengj605@163.com (S.J.); 2Department of physics, Faculty of science and technology, University of Shendi, Shendi P.O. Box 407, Sudan

**Keywords:** strain rate, 316 L austenitic stainless-steel, grain size, plastic deformation mechanisms, molecular dynamics, embedded atom method (EAM)

## Abstract

In the present study, molecular dynamics simulations were employed to investigate the effect of strain rate on the plastic deformation mechanism of nanocrystalline 316 L stainless-steel, wherein there was an average grain of 2.5–11.5 nm at room temperature. The results showed that the critical grain size was 7.7 nm. Below critical grain size, grain boundary activation was dominant (i.e., grain boundary sliding and grain rotation). Above critical grain size, dislocation activities were dominant. There was a slight effect that occurred during the plastic deformation mechanism transition from dislocation-based plasticity to grain boundaries, as a result of the stress rate on larger grain sizes. There was also a greater sensitive on the strain rate for smaller grain sizes than the larger grain sizes. We chose samples of 316 L nanocrystalline stainless-steel with mean grain sizes of 2.5, 4.1, and 9.9 nm. The values of strain rate sensitivity were 0.19, 0.22, and 0.14, respectively. These values indicated that small grain sizes in the plastic deformation mechanism, such as grain boundary sliding and grain boundary rotation, were sensitive to strain rates bigger than those of the larger grain sizes. We found that the stacking fault was formed by partial dislocation in all samples. These stacking faults were obstacles to partial dislocation emission in more sensitive stress rates. Additionally, the results showed that mechanical properties such as yield stress and flow stress increased by increasing the strain rate.

## 1. Introduction

The study of the nanocrystalline (NC) austenitic stainless-steel plastic deformation process is important due to these materials’ influence on mechanical properties during the production process [1,2]. When the alloy is plastically deformed, its shear strain is usually produced by deformation twinning and/or dislocation slip, particularly at low strain rates. Other deformation mechanisms include grain rotation, grain boundary sliding, and grain migration, but these mechanisms only become important at high temperatures, especially when the grain sizes are large. Deformation twinning is a common and significant mechanism in metals and alloys. The stacking fault energy (SFE) is a significant parameter in the plastic deformation mechanism of a face-centered cubic (fcc), where the twinning tendency of an fcc metal is largely determined by its SFE. For example, coarse-grained (CG) fcc metals with high stacking fault energies such as Ni normally deform by dislocation slip, while fcc metals with low SFE such as Ag [3] and austenitic steels [4,5] primarily deform by twinning. Deformation twinning has been extensively observed in CG austenitic steels with low SFE during severe plastic deformation (SPD). For NC fcc materials, twinning is one of the major deformation mechanisms even with high SFE. Therefore, the focus of this paper is plastic deformation on NC fcc 316 L stainless-steel.

NC materials are defined as substances that have a grain size less than 100 nm [6]. These nanocrystalline materials are required in the research field because they have excellent properties, such as high strength and hardness, compared to CG materials. However, ductility is usually significantly reduced. Ultrafine-grained materials are synthesized using two methods: nano-powder synthesis and SPD. For instance, SPD, high-pressure torsion, equal channel angular processing, accumulative roll-ponding, accumulative press-bonding, ball milling, multidirectional forging, and so on, are used to produce grain size in the nanostructure. As the grain boundaries play a significant role in defining the mechanical properties of nanomaterials, they are often in non-equilibrium states, or more rigorously, they are diffuse boundaries between dislocation cells in the nanocrystalline produced by SPD. Moreover, there are many other defects like stacking faults and twins, and these defects and non-equilibrium grain boundaries may significantly influence mechanical behavior.

The mechanical properties of nanocrystalline stainless-steel, including flow stress, yield stress, and ductility are controlled by plastic deformation mechanisms. Plastic deformations may occur through different mechanisms, such as partial dislocations formed from grain boundaries [7,8], deformation twinning [9,10,11,12,13], perfect dislocation gliding [14], grain boundary sliding [15,16], phase transformation [17,18], and grain boundary rotation [19,20]. The phase transformation is one of the most important mechanisms, where phase transformation from a face-centered cubic to a body-centered cubic (fcc→bcc) plays an important role in determining the mechanical properties of CG 316 L stainless-steel. In this study, we focused only on the deformation mechanisms that occur in NC fcc 316 L stainless-steel. Deformation twinning and dislocation slip are two competitive deformations. Interactions between deformation twinning and dislocation slip occur in the grain boundary. Dislocation slip and twinning activities enhance the strength and ductility of nanocrystalline materials. Deformation twinning has been extramaritally observed in CG metals and alloys. Deformation twinning occurs in fcc metals with low SFE.

Molecular dynamics (MD) simulation has been previously used to examine the mechanical properties and plastic deformation mechanisms of nanocrystalline metals with average grain sizes [21,22]. Moreover, MD simulation has been employed to study the strain rate impact on the plastic deformation with the different grain sizes at room temperature. Zhou et al. [23] used MD simulations to investigate the influence of grain size on NC copper, indicating that the plastic deformation mechanism transits from the dislocation-mediated plasticity to grain boundary sliding in a small grain size around 10 nm. Schiøtz et al. [24] used MD simulation to study nanocrystalline copper plastic deformation, showing that flow stress decreases when the strain rate decreases. Further, they observed a transition from dislocation-mediated plasticity to grain boundary sliding for larger grain sizes.

Since it is difficult to produce NC materials for experimental studies, early studies were conducted on NC samples created by inert gas condensation, which had no clean grain boundaries or defects, such as cracks. In this study, we used MD simulation, which has many advantages compared to other experimental researches, making it a powerful tool to study NC. MD simulations can expound atomic-level deformation mechanisms in NC material in a degree of detail that cannot be obtained via experimental studies. Moreover, it has the ability to investigate deformation mechanisms in real-time, such as twinning, stacking faults, and other faults. It can deform samples into extensive plastic strains, making it possible to explore the deformation mechanism with high dislocations.

In this study, we created 316 L of NC stainless-steel samples with a mean grain size between 2.5 to 11.5 nm. MD simulation was used to explain the effect of strain rate on these samples. Further, we discuss plastic deformation mechanisms at different strain rates.

## 2. Simulation Methods

The Voronoi construction method [25] and ATOMSK [26] software (Materials and Transformations Unit, Bât. C6, Univ. Lille 1, 59655 Villeneuve d´Ascq, France) were used to create three-dimensional (3D) polycrystalline stainless-steel samples with mean grain sizes ranging from 2.5 to 11.5 nm. A polycrystalline simulation box contained fcc iron (Fe) (200 × 200 × 200 Å^3^) and a sum of 691,554 atoms. The nickel (Ni) and chromium (Cr) elements were replacement atoms and were randomly handed out in the sample. The sample was composed of Ni-12%, Cr-17%, and Fe-71%, as shown in Figure 1a. Although other alloy elements, such as Mo, Mn, C, and N, are reported to play a crucial role in determining the mechanical properties of nanocrystalline 316 L stainless-steel, given the complexity and lack of reliable interatomic potentials of multiple elements, the current work only considers the influence of alloy elements, namely Fe, Cr, and Ni. Grain directions were randomly organized in the samples. Figure 1b,c present a sample with a mean grain size of 2.5 nm.

After the polycrystalline stainless-steel samples were created, molecular dynamics simulations (LAMMPS) progressed by the Sandia National Laboratories [27] were implemented using a large-scale atomic molecular massively parallel simulator. Periodic boundary conditions were applied in x-, y-, and z-dimensions. The embedded atom method (EAM), developed by Zhou et al. [28], was also utilized. All polycrystalline samples were minimized using the conjugate gradient algorithm to get a steady atom configuration. The simulation sample was applied at 300 K using a Nose–Hoover thermostat (Research School of Chemistry, Australian National University, G.P.O. Box 4, Canberra, ACT. 2601, Australia) [29,30] under the time steps of 2 femtoseconds. Before deformation simulation, the samples were relaxed at 300 k and a pressure of 0 bar with a time step of 2 fs for time 40 ps, measured using a Nose–Hoover thermostat and Parrinello–Rahman barostat (NPT). Tensile deformation along the x-axis is simulated at strain rates of 6 × 10^9^ and 1 × 10^10^ s^−1^ at 300 K.

After relaxation, the bulk NC 316 L austenite stainless-steel was tensile loaded and applied at room temperature. A constant strain rate (1.0 × 10^10^ s^−1^) was implemented in which the length (in the x-direction) of the bulk NC 316 L austenite stainless-steel was measured. The atomic configuration throughout the simulation was visualized using OVITO software (Institute for Materials Science, Technical University Darmstadt, Petersenstr, Darmstadt, Germany) [31]. The common-neighbor analysis technique (CNA) was used to visualize the crystal defects and local atomic structure [32]. Different types of crystal defects and atoms were marked in colors such as green for fcc structure, blue for bcc structure, red for stacking faults, and white for grain boundaries or dislocation cores.

## 3. Results and Discussion

### 3.1. Tensile Deformation

Figure 2 illustrates the tensile stress–strain curves for NC 316 L stainless-steel for various average grain sizes. Strain rate was applied to all samples ranging from 1 × 10^10^ s^−1^ to 6 × 10^9^ s^−1^, respectively. Yield stress increased with a higher strain rate. In all investigated samples, a linear correlation between stress and strain meant that stainless-steel alloys with different grain sizes had an elastic region, which can be obtained from the elastic constant by fitting the initial linear elastic region. Plastic deformation might happen in the linear elastic phase at a high strain rate. These plastic deformations occurred within the grain boundaries such as the grain boundary movement and could not be observed by any grain boundary dislocation activity in the elastic region during the deformation simulation.

For samples with a grain size above the critical grain sizes, an increase in the neck in the stress–strain curves (Figure 2e,f) can be noted, particularly for high strain rates. This can be explained by the fact that there was insufficient time for nucleated dislocations when the plastic deformation appeared through large grain sizes and high strain rates. For samples below the critical grain size (3.6 and 4.1 nm), fluctuations in the loading tensile curves at various strain rates were observed (Figure 2b,c). This may indicate that the dominant mechanism was the grain boundary movement and not the dislocations. 

Figure 3 illustrates the Hall–Petch (H–P) relationship’s transition from normal to inverse, which is dependent on the flow stress at various strain rates. In Figure 3, the extreme in the flow stress corresponds to 7.7 nm for strain rates of 1 × 10^10^ s^−1^ and decreases when strain rate decreases. The transition from a normal to inverse H–P relationship was noted in NC material via MD simulations. This transition occurred after a shift in grain boundary, which was mediated by dislocation-based plasticity. Figure 4 shows how the flow stress increased when the strain rate increased below a critical grain size (d ˂ 7.7 nm). Samples with a high flow stress had small strain rate sensitivity. The maximum flow stress was observed using MD simulations in copper NC at a mean grain size around 8–20 nm. Figure 4 illustrates flow stress dependence on the strain rate. For all samples, the flow stress increments expanded the strain rate, especially for the sample with a grain size of 7.7 nm at a strain rate of 1 × 10^10^ s^−1^. The strain rate sensitivity dependence of the different grain size was investigated in the NC stainless-steel samples. The sample with a grain size of 9.9 nm flow stress slowly increased when the strain rate increased. When the strain rate increments moved from 6 × 10^9^ s^−1^ to 1 × 10^10^ s^−1^, strain rate sensitivity increments moved from 0.122 to 0.135. For a small grain size of 2.5 nm, flow stress increments quickly expanded the strain rate. When the strain rate increased from 6 × 10^9^ s^−1^ to 1 × 10^10^ s^−1^, the volume of the strain rate sensitivity moved from 0.19 to 0.22.

### 3.2. Deformation Mechanism

To investigate the effect of strain rate on the plastic deformation of NC stainless-steel, we chose two samples: one with a mean small grain size of 3.6 nm and one with a large grain size of 9.9 nm.

For the smaller grain size, Figure 5 shows the NC stainless-steel sample with a mean grain size of 3.5 nm at strain rates of 6 × 10^9^ s^−1^ and 1 × 10^10^ s^−1^, respectively. When the applied strain was 0.075, for the lesser strain rate (6 × 10^9^ s^−1^) partial dislocations are observed emitted from grain boundaries and propagated in grains, as shown in Figure 5a. In Figure 5, partial dislocations were indicated by black arrows, while stacking faults were indicated by yellow arrows and twins indicated by blue arrows. Figure 5a shows an increase in the thickness of grain boundaries, where small grains disappeared after an increase in grain boundary thickness, merging with larger grains. This merger and the increase in the thickness grain boundaries indicated a grain rotation and grain sliding at the highest stress of 1 × 10^10^ s^−1^. Partial dislocations emitted from grain boundaries delayed emission and very few compared to those shown in Figure 5a. At the same strain (0.075; see Figure 5d), there were very few stacking faults that appeared behind the partial dislocation. When increase strains were 0.125 and 0.15, many stacking faults were observed. Stacking faults were observed in a 6 × 10^9^ s^−1^ strain rate. This compares well with the 1 × 10^10^ s^−1^ strain rate. In smaller grain sizes, the stacking fault mainly parallels in each grain; this was different from the larger mean grain sizes, where many stacking faults crisscrossed with one another at the same strain rates.

As previously mentioned, the dominate mechanism on the small grain size was the grain sliding and grain rotation. Figure 5a shows that this mechanism was more active at a strain rate of 6 × 10^9^ s^−1^. Figure 5e shows that at the strain rate of 6 × 10^9^ s^−1^, a smaller grain size appeared due to a decrease in thickness grain boundary. Some grain boundaries became narrow. For the strain rate of 1 × 10^10^ s^−1^, some grain boundaries disappeared, indicated by the long black arrow in Figure 5f. The fraction atom in the grain boundary at 1 × 10^10^ s^−1^ was greater than 6 × 10^9^ s^−1^ in the same applied strain. The plastic deformation at a lower strain rate resulted in atom relaxation within grain boundaries. Thus, the grain boundary network remained significantly unchanged.

The dominant mechanism of the small grain size was grain sliding and grain rotation, but, interestingly, the stacking fault that resulted from partial dissociation and twin activities could still be seen in the sample for each of the strain rates (Figure 5c,f). Its presence as a twin and stacking fault meant that grain boundary deformation mechanisms needed collaboration for all partial dislocation activities. Moreover, the strain rate dependency of the flow stress was mainly related to grain boundary movement mechanisms for mean grain size samples below the critical grain size (i.e., 2.5 nm). In the current study, these mechanisms included the grain boundary sliding, grain rotation, and grain boundary migration. Hence, the value of strain rate sensitivity for these grain boundary activities was 0.19 for strain rates below 1 × 10^10^ s^−1^ (Figure 4), which was smaller than the thermally activated grain boundary activities. With the increase of the strain from 0.15 to 0.2 (Figure 5c–f), the density of stacking faults increased. However, the flow stress was about steady when the strain increased (Figure 2a). In this region, we confirmed that the density of stacking faults did not affect flow stress, given that the stacking faults act as barriers to the motion of partial dislocations.

Figure 6 shows the sample of NC stainless-steel with a mean grain size of 9.9 nm at strain rates of 6 × 10^9^ s^−1^ and 1 × 10^10^ s^−1^, respectively. When the strain increased to 0.075, the partial dislocation emissions from the grain boundary mobilized in the grains (grain 1), while partial dislocations of the sample (grain 1) at 1 × 10^10^ s^−1^ and at the same strain (0.075) nucleated the grain boundaries and did not emit inside the grain (Figure 6b). The plastic deformation was mostly accommodated by grain boundary activities. Dislocation activities and deformation twinning dominated plastic deformation in the sample with an average grain size of 9.9 nm. The partial dislocation emission from the grain boundary (Shockley dislocation) is indicated by black arrows in Figure 6a at a strain rate of 6 × 10^9^ s^−1^, while impartial dislocation emission had a strain rate of 1 × 10^10^ s^−1^ at the same strain. This demonstrates that the strain rate impacted partial dislocation activities. The stacking faults were generated by increasing partial dislocations. Figure 6c,d show the stacking faults via yellow arrows at a strain of 0.125. For a strain rate of 6 × 10^9^ s^−1^, deformation twins and stacking faults were parallel in grain 1 and grain 2 and were also created by the consecutive nucleated partial dislocation at the neighbor atom planes. For a high strain rate of 1 × 10^10^ s^−1^, many stacking faults crossed with one another in most grains due to two types of partial dislocations moving in two different directions, forming intersecting stacking faults. These stacking faults hindered the movement of the partial dislocations. When the strain increased to 0.15, many deformation twins and stacking faults were observed in both strain rates. The stacking faults were formed by partial dislocation and were parallel in some grains (1 and 3). According to our simulation, we concluded that stacking faults hindered the movement of partial dislocation and increased flow stress with a higher stress rate.

When analyzing the microstructure of the grain size of 9.9 nm at different strain rates, in addition to analyzing partial dislocations occurring inside the grain, one can notice grain boundary activities, such as grain migration. Grain 4 in Figure 6a, indicated by the black dotted circle, was split into two grains. Grain boundary activities at a strain rate of 6 × 10^9^ s^−1^ were compared with a strain rate of 1 × 10^10^ s^−1^ (Figure 6c,d). Therefore, we found that the strain rate had a clear effect on grain boundary movement. At a high strain rate, the grain boundary movement was reduced, but a local strike was observed around the grain boundary. Figure 7 illustrates how the strain increased to 0.2. Moreover, Figure 7 shows how the stacking faults and twins increased. Finally, when the grain size is large, the dominant plastic deformation mechanism dislocates and deforms twins in the sample, and the directions produced by the twins are mostly perpendicular to each other. Moreover, other types of twins were formed through successive partial dislocations and interfered between two extended neighboring slip planes.

Further investigations were conducted on samples sized 2.5 nm and 9.9 nm at a strain of 0.2. We observed plastic deformation on portions of the sample. For the 2.5 nm-sized (Figure 7a,c), the dominant plastic deformation experienced grain boundary movement. However, large numbers of stacking faults and twinning were observed. For the sample with 9.9 nm average grain size (Figure 7b,d), the dominant plastic deformation experienced dislocated and deformed twinning. Many stacking faults were generated by partial dissociation movements at high strain rates and most plastic deformation was absorbed into the sample.

## 4. Conclusions

Voronoi construction was performed to create NC stainless-steel samples with mean grain sizes of 2.5 and 9.9 nm. These samples were investigated using molecular dynamics simulation to determine the effect of strain rate on NC stainless-steel plastic deformation at the strain rates of 6 × 10^9^ s^−1^ to 1 × 10^10^ s^−1^. The results of this study can be summarized as follows:The result showed that grain sliding and grain rotation dominated in small grain sizes. Additionally, dislocation activities and twins were observed for large grain sizes.Transitioning between plastic deformation mechanisms and dislocation-based plasticity to grain boundaries was observed.The influence of strain rates on small grain sizes was greater than on large grain sizes. Therefore, strain rate sensitivity increased when the grain size decreased. The stacking fault created by partial dislocation was observed in every sample.For small grain sizes or lower strain rates, the stacking fault had a partial dislocation emission. Therefore, the grain boundary movement was the most active in the small grain.For large grain sizes or high strain rates, the stacking fault did not represent any impediment to the movement of partial dislocations. Therefore, the twin was formed through the stacking fault and was spread inside the grains.Partial dislocation was observed at a lower strain rate, before it was observed at a higher strain rate, at the same applied strain.As the strain rate was increased, mechanical properties such as the yield stress and flow stress increased.

## Figures and Tables

**Figure 1 materials-13-03223-f001:**
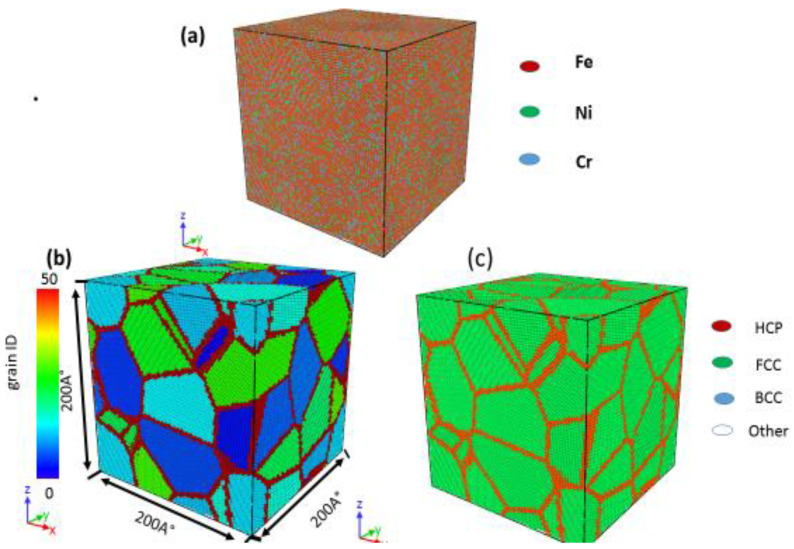
Initial configuration of the polycrystalline stainless-steel alloy with an average grain size of 2.5 nm. (**a**) Element container, (**b**) grain identity number, and (**c**) common neighbor analysis.

**Figure 2 materials-13-03223-f002:**
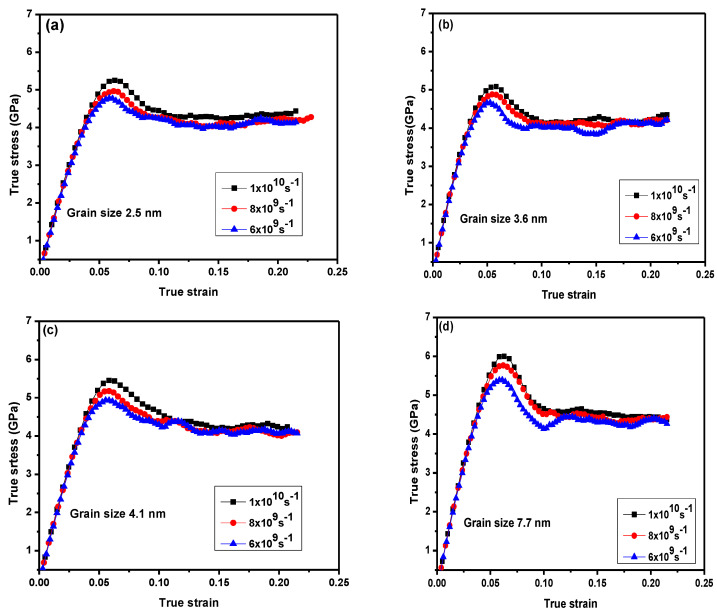
(**a**–**f**) True stress–strain curves for nanocrystalline (NC) 316 L stainless-steel with different mean grain sizes and at different strain rates (6 × 10^9^ s^−1^ to 1 × 10^10^ s^−1^).

**Figure 3 materials-13-03223-f003:**
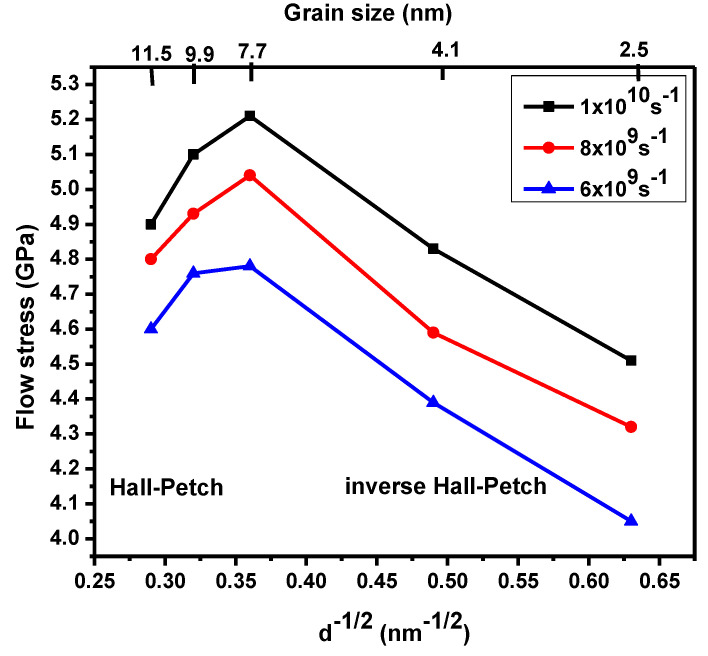
Relationship between flow stress and grain size under different strain rates.

**Figure 4 materials-13-03223-f004:**
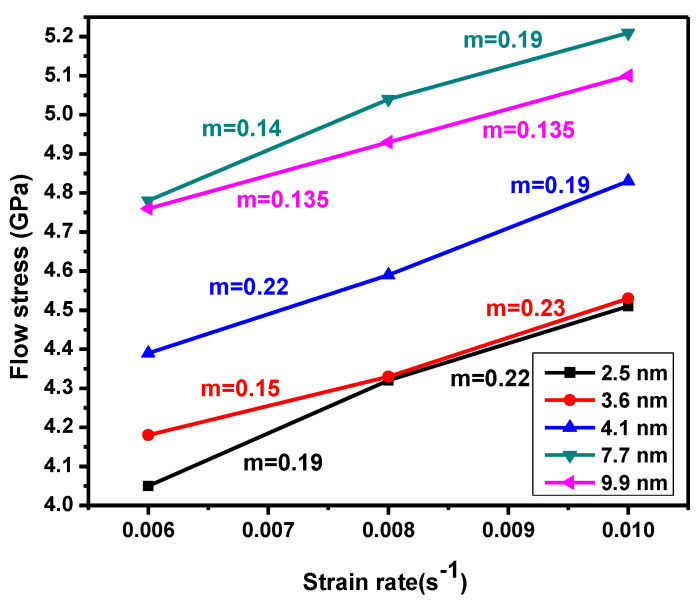
Flow stress as a function of strain rate for NC 316 L stainless-steel, with various mean grain sizes.

**Figure 5 materials-13-03223-f005:**
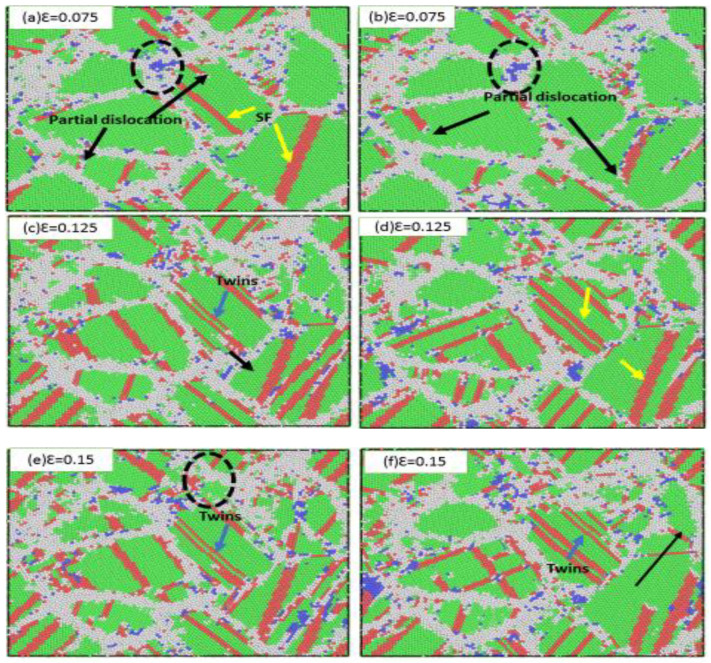
Snapshots of the simulated stainless-steel with a mean grain size of 3.6 nm, deformed at strain rates of (**a**,**c**,**e**) 6 × 10^9^ s^−1^ and (**b**,**d**,**f**) 1 × 10^10^ s^−1^. (**a**,**b**) present 0.075 true strain; (**c**,**d**) present 0.125 true strain; (**e**,**f**) present 0.15 true strain. Black arrows in (**a**,**b**) indicate partial dislocation. Yellow arrows in (**a**,**d**) indicate the stacking fault, while blue arrows indicate deformation twins.

**Figure 6 materials-13-03223-f006:**
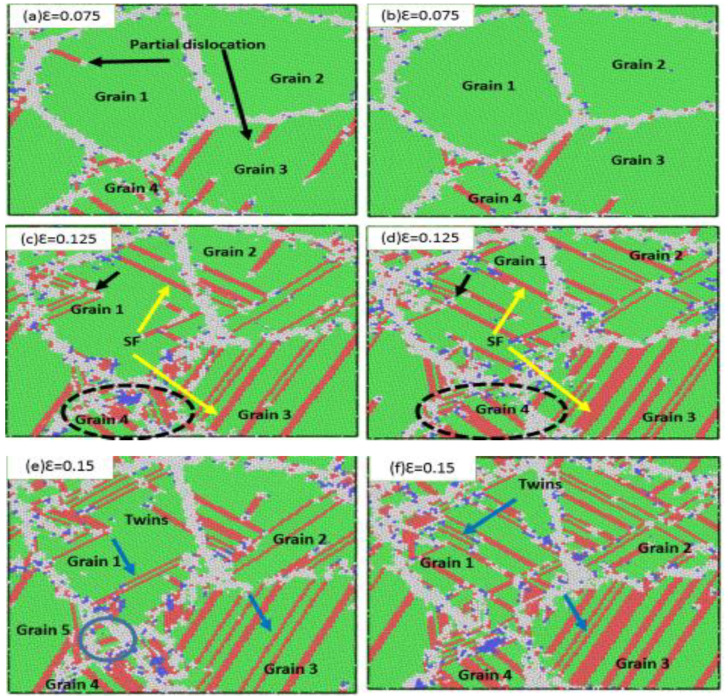
Snapshots of the simulated stainless-steel with a mean grain size of 9.9 nm, deformed at strain rates of (**a**,**c**,**e**) 6 × 10^9^ s^−1^ and (**b**,**d**,**f**) 1 × 10^10^ s^−1^. (**a**,**b**) present 0.075 true strain; (**c**,**d**) present 0.125 true strain; (**e**,**f**) present 0.15 true strain. Black arrows in (**a**,**b**) indicate partial dislocation, yellow arrows in (**c**,**d**) indicate the stacking fault, and blue arrows indicate deformation twins.

**Figure 7 materials-13-03223-f007:**
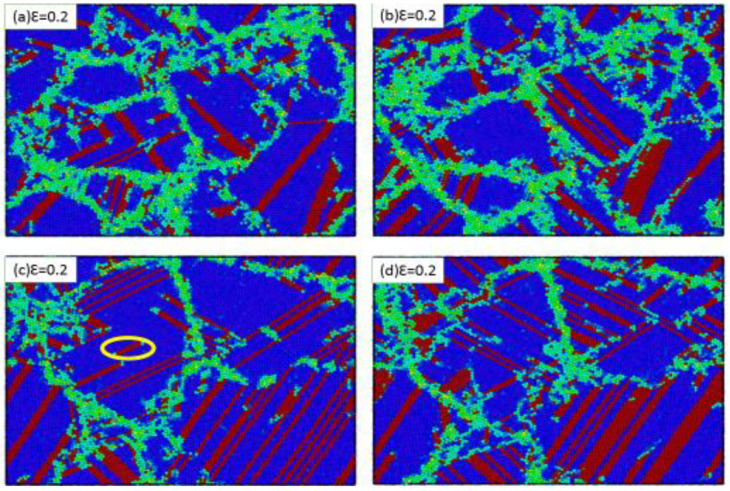
Snapshots of the simulated stainless-steel with the strain increases to 0.20. Each sample shows the strain accumulation for the sample at 6 × 10^9^ s^−1^ and 1 × 10^10^ s^−1^. This is presented in (**a**,**c**) with grain size 2.5 nm; (**b**,**d**) with grain size 9.9 nm, respectively.

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
