# Peer review of "Molecular Dynamics as a Means to Investigate Grain Size and Strain Rate Effect on Plastic Deformation of 316 L Nanocrystalline Stainless-Steel"

_materials, 2020, doi:10.3390/ma13143223_

Round 1
Reviewer 1 Report
The authors present results of molecular dynamic simulations of a nanocrystalline Cr-Ni steel subjected to tension. In principle, the results can be potentially interesting to the scientific audience dealing with investigation of fine-grained and nanocrystalline materials and their deformation mechanisms. However, there is a number of questions to the manuscript content:
1) From the "Simulation methods" section it is not clear if the authors simulate 316 steel (even 316L as follows from the title). They do not provide information on the concentration of Cr and Ni in the simulated samples. Also, it is not clear which amount of the other alloying elements are included, in particular, such important elements as C, Mn and Mo. From the description of the generated structures it seems that the simulated material does not correspond to chemical composition of the 316 steel, which typically is Fe-(0.03-0.07)C-(16-18)Cr-(up to 0.75)Si-(up to 2.0)Mn-(10-14)Ni-(2-3)Mo, wt. \%, not taking into account such minor elements as S, P or N. Variation in elements fraction even in these specified limits can alter the properties of 316 steel. The chemical composition used in the paper can rather be related to a generic Cr-Ni steel.
2) Expanding the previous remark, deformation mechanisms in austenitic steels are sensitive to chemical composition. Strain-induced phase transformations (gamma-epsilon-alpha, gamma-alpha, gamma-alpha-gamma and so on) can be an essential part of deformation mechanisms, especially, for the metastable steels. It is not clear how the paper accounts for this important issue.
3) The tensile deformation is simulated for the huge strain rates of about 10^10 s^-1 typical for explosion conditions. How such an approximation could affect the results obtained? How the neglected atomic transport mechanisms critical to grain rotation, grain boundary migration and grain boundary sliding would modify the simulated phenomena if occurred at the typical tensile strain rates by 13-14 orders of magnitude less? This is not discussed in the paper.
4) Strain-induced re-distribution of alloying elements in steels, which is often discussed recently in the literature, is also not at least mentioned in the paper.
5) The Hall-Petch plot is usually considered for the yield stress, while authors consider flow stress values. The revealed regularities, such as a breakdown in the positive Hall-Petch effect observed at a grain size of 7.7 nm, are not discussed from the viewpoint of experimental data, available in literature. As I can see, almost no experimental literature was discussed when analyzing the simulated data.
6) As a minor comment, plots in Fig. 2 could be displayed in the same scale for better visual comparison.
7) The last but not the least - the paper is written in a rather careless way. Sometimes the reader has to guess what authors did mean. From the first paragraph (lines 30-36) - "...due to their on the influence of mechanical properties..." - not grammatically correct; "Deformation mechanisms such as grain rotation, grain boundary sliding, and migration." - the sentence is just incomplete; "deformation twinning and dislocation are the diffuse and important mechanisms" - "diffuse" is not a correct term here, and dislocation - is not a mechanism. As a result even the first paragraph reads as not quite consistent and sound. And this can be related to the entire introduction and partially to the following sections. Above all, some abbreviations are introduced not at the first mentioning (NC) and then again "nanocrystalline" is often used. Some abbreviations are never used after their introduction (SPD, ARB, ECAP, MDF etc). "Grain boundary movement" - does it correspond to grain boundary migration? So the paper needs in extensive language and style editing.
To sum up, the manuscript has to be totally re-written to be considered for publication in the Materials journal.
Author Response
Response to Reviewer 1 Comments

Reviewer 2 Report
Dear Authors,
Congratulations on a sound and complex study, yet English language and style needs to be improved. I had serious problems in understanding whole phrases.
An thorough check in mandatory.
Line 25 - "properties such as elastic modulus, yield ..." you state that the elastic modulus increases with increasing the strain rate, yet on your stress - strain curves this increase cannot be observed.
In line 110 you make a reference to figure 3, but it is actually figure 2.
Line 128-129 you state "in the neck in the stress-strain curves..." - I fail to understand what you mean. Necking occurs in the test sample at higher strains.
Line 197 - you state "the value of m for these...". What is the meaning of m? I did not find it in text.
Please address these requirements.
My best regards.
Author Response
Response to Reviewer 2 Comments

Reviewer 3 Report
The topic of the manuscript entitled “Grain size and strain rate effect on plastic deformation of nanocrystalline 316L stainless steel investigated by molecular dynamics” falls within the scope of Materials. The paper contains interesting numerical results and corresponding analyses. It is of sufficient scientific interest and has originality in its technical content to merit publication. The authors have cited the relevant literature. Methods and interpretations of results are correct and novel. The issues were well presented. In terms of content, the analysis does not raise any objections. The arrangement of work maintains substantive continuity and constitutes a logical whole.
However, the manuscript is not suitable for publication in its present form. This paper requires minor corrections (mainly editorial).
Comments and remarks are presented below.
Figure 1 should be placed in chapter two after the first two paragraphs of this chapter. Then there will be no empty space on the second page of the article.
Citing several works in square brackets, use a comma instead of a dot.
References should be made in accordance with the Instructions for authors and template.
Authors should consider different arrangement of figures in the paper so that there are no empty spaces at the end of the manuscript pages.
The conclusions should be presented in points.
The value of labor significantly reduces the lack of any experimental verification.
Author Response
Response to Reviewer 3 Comments

Round 2
Reviewer 1 Report
I appreciate the responses to the reviewer's comments done by the authors and the corresponding paper modification. However, few issues were not fixed.
1) Point 1. I agree with authors that simulation of a multi-component steel is a laborious and complicated procedure. So the effect only of Cr and Ni is already worthy to be investigated. However, with only 17% of Cr and 12% of Ni this steel does not correspond to 316L chemical composition, which has quite particular combination of elements. These fractions of Ni and Cr could also correspond to commercial 321 or 347 steel specifications if we exclude Mo from consideration, for example. And these steels could have different deformation behaviour especially from the viewpoint of phase transformations to be studied in future works. I guess, a general Cr-Ni steel term can be used in the title.
1) Authors reply:
"Point 6: As a minor comment, plots in Fig. 2 could be displayed in the same scale for better visual comparison.
Fig. 2 was displayed in the same scale for better visual comparison"
However, in the attached pdf file, the figures 2(a-f) all have different scale along "true stress" axis. I propose to set the max value for this axis to 6.5 or 7 GPa for the above mentioned purpose.
2) Point 7. Authors reply that they corrected English and accuracy of the sentences. However,
- abbreviations such as "MDF" or "ARB" and so on (see comment 7 from the first reviewer's report) were used only once and never used after that - and this was not corrected despite the authors reply.
- "SPD" is still abbreviated two times.
- "These mechanisms are diffuse" (page 2) - still not corrected.
- "As the grain boundaries play a significant role in defining the mechanical properties of nanomaterials, they are often not parallel in the nanocrystalline produced with SPD." (page 2) - why grain boundaries should be parallel? Nanocrystalline what?
- "including flow yield stress" - flow or yield stress?
- "in play an important role" - sounds strange
- "Partial and perfect slip deformation twinning and dislocation are two
competitive deformations." - dislocation is a defect, not a deformation - "Interactions between deformation twinning and dislocation occur in the grain boundary" - twinning is a process, dislocation is an object...
- "with low stacking fault." - with low stacking fault energy, probably?
And so on, so on, so on... This is just the page 2. So I am afraid to say that the English and a writing style still need in improving. If the manuscript has been indeed edited by MDPI as stated by authors, then it has to be re-edited with the help of a language specialist with experience in material science.
Author Response
Response to Reviewer 1 Comments
